



# The cost function of the data fusion process and its application

Simone Ceccherini, Nicola Zoppetti, Bruno Carli, Ugo Cortesi, Samuele Del Bianco, and Cecilia Tirelli

[1]Istituto di Fisica Applicata "Nello Carrara" del Consiglio Nazionale delle Ricerche, Via Madonna del Piano 10, 50019 Sesto Fiorentino, Italy

*Correspondence to*: Simone Ceccherini (S.Ceccherini@ifac.cnr.it)

**Abstract.** When the complete data fusion method is used to fuse inconsistent measurements, it is necessary to add to the measurement covariance matrix of each fusing profile a covariance matrix that takes into account the inconsistencies. A realistic estimate of these inconsistency covariance matrices is required for effectual fused products. We evaluate the possibility of assisting the estimate of the inconsistency covariance matrices using the value of the cost function minimized in

the complete data fusion. The analytical expressions of expected value and variance of the cost function are derived. Modelling the inconsistency covariance matrix with one parameter, we determine the value of the parameter that makes the reduced cost function equal to its expected value and use the variance to assign an error to this determination. The quality of the inconsistency covariance matrix determined in this way is tested for simulated measurements of ozone profiles obtained in the thermal infrared in the framework of the Sentinel 4 mission of the Copernicus programme. As expected, the method requires

a sufficient statistics and poor results are obtained when a small numbers of profiles are being fused together, but very good results are obtained when the fusion involves a large number of profiles.

## 1 Introduction

Vertical profiles of atmospheric variables are often obtained with the inversion of remote sensing observations performed by instruments operating on space-borne and airborne platforms, as well as from ground-based stations. When the same portion

(or nearby portions) of atmosphere is measured more times by the same instrument or by different instruments the measurements can be combined in order to obtain a single vertical profile of improved quality with respect to that of the profiles retrieved from the single observation. The simultaneous retrieval from several observations is considered the most comprehensive way to combine different measurements of the same quantity (Aires et al., 2012), however, recently a new method, referred to as *Complete Data Fusion* (CDF) (Ceccherini et al., 2015), was proposed that, with simpler implementation

requirements, provides products of quality equivalent to that of the simultaneous retrieval products.

The CDF method was proved (Ceccherini, 2016) to be equivalent to the measurement space solution data fusion method (Ceccherini et al., 2009) and the latter was successfully applied to the data fusion of MIPAS-ENVISAT and IASI-METOP measurements (Ceccherini et al., 2010a; Ceccherini et al., 2010b) and of MIPAS-STR and MARSCHALS measurements (Cortesi et al., 2016).

On the other hand, as highlighted in Ceccherini et al. (2018), the CDF provides poor results when applied to inconsistent measurements. Three causes of inconsistency are possible:

  i)    the profiles to be fused (in the following referred to as *fusing profiles*) are represented on different vertical grids

  ii)    a variability is present in the observed species and the fusing profiles refer to different times and space locations

  iii)    the fusing profiles are affected by different forward model errors.

These inconsistencies were addressed in Ceccherini et al. (2018), but some problems remain open. The inconsistency of case i) was solved adding to the measurement covariance matrix (CM) of each fusing profile an interpolation CM, which is built using the grids of the fusing profiles and by the a priori CM. The inconsistency of case ii) was solved adding to the measurement CM of each fusing profile a *coincidence CM*, which describes the variability of the observed species in the field of the observations. Regarding the inconsistency of case iii) it was suggested to follow an approach similar to cases i) and ii) adding




to the measurement CM of each fusing profile a CM describing the forward model errors due, for example, to approximations in the model and uncertainties in atmospheric and instrumental parameters. The problem that remains open is the realistic estimate of these inconsistency CMs, which are otherwise determined on the basis of some educated guesses.

The value of the cost function, which is minimized in the fusion process, depends on the inconsistency CMs and can be used

to establish some constraints on their amplitude. The goal of this paper is to determine the expected value of the cost function and to use this expectation to build a procedure for the improvement of our educated guess of the inconsistency CMs.

In order to assess its advantages we apply this procedure to simulated measurements of ozone profiles obtained in the thermal infrared in the framework of the Sentinel 4 mission (ESA, 2017) of the Copernicus programme (https://sentinel.esa.int/web/sentinel/missions).

The paper is organized as follows: in Section 2, we recall the formulas of the CDF method in order to establish the formalism used in the subsequent sections. In Sections 3, we describe the properties of the cost function and in particular determine the expected value and the variance of the cost function distribution. In Section 4, we describe the method that estimates the inconsistency CMs using the expected value of the cost function, apply it to the determination of the coincidence CM and assess the cases in which it provides useful information. Conclusions are drawn in Section 5.

## 2 The CDF method

Let us assume to have $N$ independent measurements of the vertical profile of an atmospheric target referred to the same space-time location. Performing the retrieval of the $N$ measurements with the optimal estimation method (Rodgers, 2000), we obtain $N$ vectors $\hat{\mathbf{x}}_i$ ($i$=1, 2, …, $N$), here assumed to be estimates of the same profile made on a common vertical grid. The vectors $\hat{\mathbf{x}}_i$ are characterized by the CMs $\mathbf{S}_i$ and the averaging kernel matrices (AKMs) $\mathbf{A}_i$ (Ceccherini et al., 2003; Ceccherini and Ridolfi,

2010; Rodgers, 2000). The CMs $\mathbf{S}_i$ are each defined as $<\boldsymbol{\sigma}_i\boldsymbol{\sigma}_i^T>$, where the vector $\boldsymbol{\sigma}_i$ contains the errors on the vertical profile obtained propagating the errors of the observations through the retrieval process, the superscript $T$ indicates the transpose of the vector and the symbol $<…>$ indicates the statistical expected value.

The fused profile $\mathbf{x}_f$ of these $N$ measurements, as provided by the CDF method, is obtained minimizing the following cost function (see Ceccherini et al., 2015):

$$c\left(\mathbf{x}\right) = \sum_{i=1}^{N}\left(\boldsymbol{\alpha}_i - \mathbf{A}_i\mathbf{x}\right)^T \mathbf{S}_i^{-1}\left(\boldsymbol{\alpha}_i - \mathbf{A}_i\mathbf{x}\right) + \left(\mathbf{x} - \mathbf{x}_a\right)^T \mathbf{S}_a^{-1}\left(\mathbf{x} - \mathbf{x}_a\right),$$ (1)

where $\mathbf{x}_a$ and $\mathbf{S}_a$ are the a priori profile and its CM that are used to constrain the data fusion and

$$\boldsymbol{\alpha}_i \equiv \hat{\mathbf{x}}_i - \left(\mathbf{I} - \mathbf{A}_i\right)\mathbf{x}_{ai}$$ (2)

is a modified fusing profile with $\mathbf{x}_{ai}$ the a priori profile used in the $i$-th retrieval and $\mathbf{I}$ the identity matrix.

It is possible to verify that the modified fusing profile of Eq. (2) is also a measurement of the true profile $\mathbf{x}_t$ obtained using the averaging kernels:

$$\boldsymbol{\alpha}_i = \mathbf{A}_i\mathbf{x}_t + \boldsymbol{\sigma}_i$$ (3)

This measurement does not depend on the a-priori profile and has the same CM as the fusing profile.

The CDF solution $\mathbf{x}_f$ is the profile that corresponds to the minimum of $c(\mathbf{x})$ and is obtained with the following equation:

$$\mathbf{x}_f = \left(\mathbf{F} + \mathbf{S}_a^{-1}\right)^{-1}\left(\sum_{i=1}^{N}\mathbf{A}_i^T\mathbf{S}_i^{-1}\boldsymbol{\alpha}_i + \mathbf{S}_a^{-1}\mathbf{x}_a\right),$$ (4)

where we have introduced the matrix

$$\mathbf{F} = \sum_{i=1}^{N}\mathbf{A}_i^T\mathbf{S}_i^{-1}\mathbf{A}_i,$$ (5)



which is the Fisher information matrix (Ceccherini et al., 2012; Fisher, 1935) of the fused profile, equal to the sum of the Fisher information matrices of the fusing measurements. As indicated by the name, this matrix fully characterizes the information content of each measurement.

The fused profile is characterized by the following CM and AKM:

$$\mathbf{S}_f = \left(\mathbf{F} + \mathbf{S}_a^{-1}\right)^{-1} \mathbf{F} \left(\mathbf{F} + \mathbf{S}_a^{-1}\right)^{-1} \tag{6}$$

$$\mathbf{A}_f = \left(\mathbf{F} + \mathbf{S}_a^{-1}\right)^{-1} \mathbf{F} . \tag{7}$$

When the fusing profiles $\hat{\mathbf{x}}_i$ are represented on different vertical grids, it is necessary to perform a resampling of the AKMs (Calisesi et al., 2005) which defines new $\mathbf{A}_i'$ with their second index equal to that of the common fusion grid. Following Ceccherini et al. (2016), we define such a transformation as follows:

$$\mathbf{A}_i' = \mathbf{A}_i \mathbf{R}_i , \tag{8}$$

where $\mathbf{R}_i$ are the generalized inverse matrices of the interpolation matrices $\mathbf{H}_i$, which interpolate the fusing profiles on the fusion grid.

In general, in order to account for interpolation, coincidence and forward model errors, the CDF formula can be modified (Ceccherini et al., 2018) by replacing $\boldsymbol{\alpha}_i$ with

$$\tilde{\boldsymbol{\alpha}}_i = \boldsymbol{\alpha}_i - \mathbf{A}_i \left(\mathbf{C}^{(i)} - \mathbf{R}_i \mathbf{C}^{(f)}\right) \mathbf{x}_a , \tag{9}$$

where $\mathbf{C}^{(i)}$ and $\mathbf{C}^{(f)}$ are the sampling matrices that select the grids ($i$) and the grid ($f$), respectively, from a fine grid, that includes all the levels of the fusion grid ($f$) and of the $N$ grids ($i$), and $\mathbf{S}_i$ with

$$\tilde{\mathbf{S}}_i = \mathbf{S}_i + \mathbf{S}_{i,\text{int}} + \mathbf{S}_{i,\text{coin}} + \mathbf{S}_{i,\text{other}} , \tag{10}$$

where $\mathbf{S}_{i,\text{int}}$, $\mathbf{S}_{i,\text{coin}}$ and $\mathbf{S}_{i,\text{other}}$ are the CMs associated to the interpolation error, to the coincidence error and to the forward model

errors.

The CM associated to the interpolation error is given by

$$\mathbf{S}_{i,\text{int}} = \mathbf{A}_i \left(\mathbf{C}^{(i)} - \mathbf{R}_i \mathbf{C}^{(f)}\right) \mathbf{S}_a \left(\mathbf{C}^{(i)} - \mathbf{R}_i \mathbf{C}^{(f)}\right)^T \mathbf{A}_i^T . \tag{11}$$

The CM associated to the coincidence error is given by

$$\mathbf{S}_{i,\text{coin}} = \mathbf{A}_i \mathbf{C}^{(i)} \mathbf{S}_{\text{coin}} \mathbf{C}^{(i)T} \mathbf{A}_i^T , \tag{12}$$

where $\mathbf{S}_{\text{coin}}$ is the CM describing the variability of the true profiles of the fusing measurements.

The CM associated to the forward model errors is given by (Rodgers, 2000):

$$\mathbf{S}_{i,\text{other}} = \mathbf{G}_i \mathbf{S}_{i,\text{FM}} \mathbf{G}_i^T , \tag{13}$$

where $\mathbf{G}$ is the *gain matrix*, which includes the derivatives of the retrieved profile with respect to the observations and $\mathbf{S}_{i,\text{FM}}$ is

the CM describing the forward model errors due, for example, to approximations in the model and uncertainties in atmospheric and instrumental parameters.

### 3 The cost function

In this Section, the expected value and the variance of the cost function are derived. In order to keep the formalism as simple as possible we deal with the cost function given in Eq. (1), where the treatment of inconsistency errors is not included.

However, since the inconsistency errors only modify the CMs and the vectors $\boldsymbol{\alpha}_i$ and do not affect the fusion formula, the results obtained in this Section are valid in the general case.





Once that the fused profile $\mathbf{x}_f$ is calculated from Eq. (4) we can substitute it in Eq. (1) in order to obtain $c\left(\mathbf{x}_f\right) \equiv c^{\min}$ , that

is the minimum value of the cost function. Because of measurement errors, $c^{\min}$ does not have a definite value, but assumes

values according to a probability distribution. The properties of this probability distribution (in the following referred to as

*cost function distribution*) are considered and in particular we determine the expected value and the variance of the distribution.

In order to calculate these quantities we have to make explicit the errors $\boldsymbol{\sigma}_i$ in the expression of $c^{\min}$ , see next Section. We

assume that the errors $\boldsymbol{\sigma}_i$ are normally distributed with expected values equal to zero, have CMs equal to $\mathbf{S}_i$ and are uncorrelated

for different measurements.

### 3.1 The dependence of the cost function on the measurement errors

Substituting in Eq. (4) the expression of $\boldsymbol{\alpha}_i$ given by Eq. (3) and using Eq. (7) we obtain the following expression for $\mathbf{x}_f$:

$$\mathbf{x}_f = \mathbf{A}_f \mathbf{x}_t + \left(\mathbf{I} - \mathbf{A}_f\right)\mathbf{x}_a + \boldsymbol{\sigma}_f ,$$ (14)

where $\boldsymbol{\sigma}_f$ is the error on $\mathbf{x}_f$ given by:

$$\boldsymbol{\sigma}_f = \left(\mathbf{F} + \mathbf{S}_a^{-1}\right)^{-1} \sum_{i=1}^{N} \mathbf{A}_i^T \mathbf{S}_i^{-1} \boldsymbol{\sigma}_i$$ (15)

and characterized by the CM $\mathbf{S}_f = <\boldsymbol{\sigma}_f \boldsymbol{\sigma}_f^T>$ given in Eq. (6).

Substituting in Eq. (1) the expression of $\boldsymbol{\alpha}_i$ given by Eq. (3) and $\mathbf{x}$ with the expression of $\mathbf{x}_f$ given by Eq. (14), we obtain the

expression of $c^{\min}(\boldsymbol{\sigma}_i)$ as a function of the measurement errors:

$$c^{\min}\left(\boldsymbol{\sigma}_i\right) = \sum_{i=1}^{N} \left[\boldsymbol{\sigma}_i - \mathbf{A}_i \boldsymbol{\sigma}_f + \mathbf{A}_i\left(\mathbf{I} - \mathbf{A}_f\right)\left(\mathbf{x}_t - \mathbf{x}_a\right)\right]^T \mathbf{S}_i^{-1} \left[\boldsymbol{\sigma}_i - \mathbf{A}_i \boldsymbol{\sigma}_f + \mathbf{A}_i\left(\mathbf{I} - \mathbf{A}_f\right)\left(\mathbf{x}_t - \mathbf{x}_a\right)\right]$$
$$+ \left[\boldsymbol{\sigma}_f + \mathbf{A}_f\left(\mathbf{x}_t - \mathbf{x}_a\right)\right]^T \mathbf{S}_a^{-1} \left[\boldsymbol{\sigma}_f + \mathbf{A}_f\left(\mathbf{x}_t - \mathbf{x}_a\right)\right]$$ (16)

where $\boldsymbol{\sigma}_f$ is a linear function of $\boldsymbol{\sigma}_i$ expressed by Eq. (15).

Eq. (16) contains several matrix products, which produce several terms; we can rearrange these terms in the following way:

$$c^{\min}\left(\boldsymbol{\sigma}_i\right) = c^{\min}_0 + c^{\min}_1\left(\boldsymbol{\sigma}_i\right) + c^{\min}_2\left(\boldsymbol{\sigma}_i\right) ,$$ (17)

where $c^{\min}_0$ is independent of the errors, $c^{\min}_1(\boldsymbol{\sigma}_i)$ is linear in the errors and $c^{\min}_2(\boldsymbol{\sigma}_i)$ is quadratic in the errors.

In the case of the term independent of the errors, performing algebraic operations and using Eqs. (5) and (7), we obtain:

$$c^{\min}_0 = \sum_{i=1}^{N} \left[\mathbf{A}_i\left(\mathbf{I} - \mathbf{A}_f\right)\left(\mathbf{x}_t - \mathbf{x}_a\right)\right]^T \mathbf{S}_i^{-1} \left[\mathbf{A}_i\left(\mathbf{I} - \mathbf{A}_f\right)\left(\mathbf{x}_t - \mathbf{x}_a\right)\right] +$$
$$+ \left[\mathbf{A}_f\left(\mathbf{x}_t - \mathbf{x}_a\right)\right]^T \mathbf{S}_a^{-1} \left[\mathbf{A}_f\left(\mathbf{x}_t - \mathbf{x}_a\right)\right] =$$
$$= \left(\mathbf{x}_t - \mathbf{x}_a\right)^T \mathbf{S}_a^{-1} \mathbf{A}_f\left(\mathbf{x}_t - \mathbf{x}_a\right) = tr\left[\left(\mathbf{x}_t - \mathbf{x}_a\right)\left(\mathbf{x}_t - \mathbf{x}_a\right)^T \mathbf{S}_a^{-1} \mathbf{A}_f\right]$$ (18)

where $tr[]$ identifies the trace of the matrix and we have used the relation for the trace of a product of two matrices

$tr[\mathbf{CD}]=tr[\mathbf{DC}]$ when $\mathbf{D}$ and $\mathbf{C}^T$ have the same shape.

In the case of the term linear in the errors, performing algebraic operations and using Eqs. (5), (7) and (15), we obtain:

$$c^{\min}_1\left(\boldsymbol{\sigma}_i\right) = 2\sum_{i=1}^{N} \left[\mathbf{A}_i\left(\mathbf{I} - \mathbf{A}_f\right)\left(\mathbf{x}_t - \mathbf{x}_a\right)\right]^T \mathbf{S}_i^{-1} \left[\boldsymbol{\sigma}_i - \mathbf{A}_i \boldsymbol{\sigma}_f\right] + 2\left[\mathbf{A}_f\left(\mathbf{x}_t - \mathbf{x}_a\right)\right]^T \mathbf{S}_a^{-1} \boldsymbol{\sigma}_f =$$
$$= 2\left(\mathbf{x}_t - \mathbf{x}_a\right)^T \mathbf{S}_a^{-1} \boldsymbol{\sigma}_f$$ (19)

In the case of the term quadratic in the errors, performing algebraic operations and using Eqs. (5) and (15), we obtain:



$$c^{\min}{}_2\left(\boldsymbol{\sigma}_i\right) = \sum_{i=1}^{N}\left(\boldsymbol{\sigma}_i - \mathbf{A}_i\boldsymbol{\sigma}_f\right)^T \mathbf{S}_i^{-1}\left(\boldsymbol{\sigma}_i - \mathbf{A}_i\boldsymbol{\sigma}_f\right) + \boldsymbol{\sigma}_f{}^T \mathbf{S}_a^{-1}\boldsymbol{\sigma}_f =$$

$$= \sum_{i=1}^{N}\boldsymbol{\sigma}_i{}^T \mathbf{S}_i^{-1}\boldsymbol{\sigma}_i - \boldsymbol{\sigma}_f{}^T\left(\mathbf{F} + \mathbf{S}_a^{-1}\right)\boldsymbol{\sigma}_f \qquad (20)$$

From Eqs (17-20) we obtain that the full expression of $c^{\min}(\boldsymbol{\sigma}_i)$, arranged as a function of the errors, is:

$$c^{\min}\left(\boldsymbol{\sigma}_i\right) = tr\left[\left(\mathbf{x}_t - \mathbf{x}_a\right)\left(\mathbf{x}_t - \mathbf{x}_a\right)^T \mathbf{S}_a^{-1}\mathbf{A}_f\right] + 2\left(\mathbf{x}_t - \mathbf{x}_a\right)^T \mathbf{S}_a^{-1}\boldsymbol{\sigma}_f + \sum_{i=1}^{N}\boldsymbol{\sigma}_i{}^T \mathbf{S}_i^{-1}\boldsymbol{\sigma}_i - \boldsymbol{\sigma}_f{}^T\left(\mathbf{F} + \mathbf{S}_a^{-1}\right)\boldsymbol{\sigma}_f, \qquad (21)$$

where $\boldsymbol{\sigma}_f$ is a function of $\boldsymbol{\sigma}_i$ according to Eq. (15).

### 3.2 Expected value of the cost function

The expected value of the cost function is equal to the summation of the expected values of its three terms. Since $c^{\min}{}_0$ is independent of the errors, its expected value coincides with its constant value. The expected value of $c^{\min}{}_1(\boldsymbol{\sigma}_i)$ is zero because this term is linear in $\boldsymbol{\sigma}_i$, and the expected values of $\boldsymbol{\sigma}_i$ are equal to zero. Therefore, we need to calculate only the expected value of $c^{\min}{}_2(\boldsymbol{\sigma}_i)$:

$$\left\langle c^{\min}{}_2\left(\boldsymbol{\sigma}_i\right)\right\rangle = \sum_{i=1}^{N}\left\langle \boldsymbol{\sigma}_i{}^T \mathbf{S}_i^{-1}\boldsymbol{\sigma}_i\right\rangle - \left\langle \boldsymbol{\sigma}_f{}^T\left(\mathbf{F} + \mathbf{S}_a^{-1}\right)\boldsymbol{\sigma}_f\right\rangle = \sum_{i=1}^{N}tr\left(\left\langle \boldsymbol{\sigma}_i\boldsymbol{\sigma}_i{}^T\right\rangle \mathbf{S}_i^{-1}\right) - tr\left(\left\langle \boldsymbol{\sigma}_f\boldsymbol{\sigma}_f{}^T\right\rangle\left(\mathbf{F} + \mathbf{S}_a^{-1}\right)\right) =$$

$$= \sum_{i=1}^{N}tr\left(\mathbf{I}_i\right) - tr\left(\mathbf{S}_f\left(\mathbf{F} + \mathbf{S}_a^{-1}\right)\right) = \sum_{i=1}^{N}n_i - tr\left(\mathbf{A}_f\right) \qquad (22)$$

where Eqs. (6) and (7) have been used and $n_i$ is the number of eigenvalues different from zero of $\mathbf{S}_i^{-1}$ rather than the number of its diagonal elements. When $\mathbf{S}_i$ is singular (or near singular) the inversion is performed by means of the generalized inverse (Kalman, 1976), and therefore, $\mathbf{S}_i^{-1}$ may have some eigenvalues equal to zero.

Finally, the expected value of the cost function is given by:

$$\left\langle c^{\min}\left(\boldsymbol{\sigma}_i\right)\right\rangle = \sum_{i=1}^{N}n_i - tr\left(\mathbf{A}_f\right) + tr\left[\left(\mathbf{x}_t - \mathbf{x}_a\right)\left(\mathbf{x}_t - \mathbf{x}_a\right)^T \mathbf{S}_a^{-1}\mathbf{A}_f\right]. \qquad (23)$$

Recalling that the trace of the AKM represents the number of degrees of freedom (DOFs), which is the number of independent parameters actually determined by the analysis (Rodgers, 2000), we see that the expected value of the cost function is equal to: a first term that counts the number of available measurements minus a second term that is the number of DOFs plus a third term that depends on the difference between the a priori profile and the true profile.

### 3.3 Variance of the cost function

Using Eq. (21) it is possible to calculate the expression of the variance of the cost function. For those interested, the lengthy calculation is reported in Appendix A. The result is:

$$\text{var}\left[c^{\min}\left(\boldsymbol{\sigma}_i\right)\right] = 2\sum_{i=1}^{N}n_i - 4tr\left(\mathbf{A}_f\right) + 2tr\left(\mathbf{A}_f{}^2\right) + 4tr\left[\left(\mathbf{x}_t - \mathbf{x}_a\right)\left(\mathbf{x}_t - \mathbf{x}_a\right)^T \mathbf{S}_a^{-1}\mathbf{A}_f\left(\mathbf{I} - \mathbf{A}_f\right)\right]. \qquad (24)$$

Eqs. (23) and (24) provide new relationships that make possible to calculate the expected value and the variance of the cost function minimized in the CDF.

A particular case is that in which we take a priori errors going to infinity (unconstrained case). In that case $\mathbf{S}_a^{-1}$ tends to the null matrix and $\mathbf{A}_f$ coincides with the identity matrix, therefore, we obtain:

$$\left\langle c^{\min}\left(\boldsymbol{\sigma}_i\right)\right\rangle_{\mathbf{S}_a^{-1}\to 0} = \sum_{i=1}^{N}n_i - n \qquad (25)$$





$$\mathrm{var}\left[c^{\min}\left(\boldsymbol{\sigma}_i\right)\right]_{\mathbf{S}_a^{-1}\to 0} = 2\left(\sum_{i=1}^{N} n_i - n\right), \tag{26}$$

where $n$ is the number of levels of the fused profile. As expected, Eqs (25-26) are equal to the expected value and the variance of the chi-square distribution.

More generally, we notice that the third term of Eq. (23) and the fourth term of Eq. (24), which are only present when a constraint is used for the calculation of the fused profile, are a very small correction whenever mild constraints are used.

## 3.4 Reduced cost function

It is useful to introduce the *reduced cost function* defined as the ratio between the cost function and the expected value of the cost function:

$$c_r\left(\mathbf{x}\right) = \frac{c\left(\mathbf{x}\right)}{\left\langle c^{\min}\left(\boldsymbol{\sigma}_i\right)\right\rangle} \tag{27}$$

with an expected value equal to 1.

Accordingly, the variance of the reduced cost function is equal to:

$$\mathrm{var}\left[c_r{}^{\min}\left(\boldsymbol{\sigma}_i\right)\right] = \frac{\mathrm{var}\left[c^{\min}\left(\boldsymbol{\sigma}_i\right)\right]}{\left\langle c^{\min}\left(\boldsymbol{\sigma}_i\right)\right\rangle^2}. \tag{28}$$

## 4 Application

### 4.1 Method to estimate the inconsistency CMs

When the correct CMs are used, the reduced cost function is bound to be equal to one within the variability determined by its variance. In turn using the expected value of the reduced cost function as a constraint we can tune the values of the CMs that characterize the inconsistencies of the fusing profiles, in particular either the CM $\mathbf{S}_{\mathrm{coin}}$ describing the variability of the true profiles of the fusing measurements or the CMs $\mathbf{S}_{i,\mathrm{FM}}$ describing the forward model errors. Of course the reduced cost function is a single constraint, furthermore limited by the uncertainty introduced by its variance, and can only be used to determine one parameter of the inconsistency CMs. However, if the same unknown CM is involved in several fusion processes a more elaborate determination of the CM may also be considered. In the following we consider the simple case in which the inconsistency CM is parametrized with a single parameter.

### 4.1.1 Estimate of the $k$ parameter

If the inconsistency CM is written as $k\boldsymbol{\Sigma}$, where $k$ is a multiplicative parameter and $\boldsymbol{\Sigma}$ is an assumed CM that describes the inconsistency error, the value of the $k$ parameter can be determined imposing that the reduced cost function is equal to one:

$$c_r\left[\mathbf{x}_f\left(k\right), k\right] = 1. \tag{29}$$

Since $c_r[\mathbf{x}_f(k),k]$ is a monotonic decreasing function of $k$, the value of $k$ satisfying Eq. (29) can be easily found numerically.

### 4.1.2 Estimate of the error of the $k$ parameter

The variance of the reduced cost function determines an error $\Delta k$ on the value of the parameter $k$ that is given by the following expression:



$$\Delta k = \frac{\sqrt{\mathrm{var}\left\{c_r\left[\mathbf{x}_f\left(k\right),k\right]\right\}}}{\dfrac{dc_r\left[\mathbf{x}_f\left(k\right),k\right]}{dk}}\ . \tag{30}$$

The determination of $k$ and $\Delta k$ by means of Eqs. (29) and (30) requires the calculation of $<c^{\mathrm{min}}(\boldsymbol{\sigma}_i)>$ and $\mathrm{var}[c^{\mathrm{min}}(\boldsymbol{\sigma}_i)]$ by means of Eqs. (23) and (24), which depend on the true profile. Since the true profile is unknown, in the following analysis we replace the true profile with the fused profile, which is its best estimate.

**4.2 The determination of the coincidence CM in the case of simulated ozone data**

**4.2.1 Simulated data**

The use of CDF will be particularly relevant for the analysis of the future atmospheric Sentinel missions of the Copernicus programme (https://sentinel.esa.int/web/sentinel/missions). The amount of data that will be available from these missions will pose technical challenges to most applications and the CDF can be used to reduce the number of products while maintaining the information content of the full datasets. In general, we have a good understanding of the average geographical variability

of the observed products and a reasonable assumption can be made of the $\mathbf{S}_{\mathrm{coin}}$ that is used for the data fusion, but local fluctuations may also have significant effects. Therefore, the possibility of using a scalar $k$ which takes into account the local fluctuations may provide for these data an important improvement. For this reason, simulated data of the Sentinel 4 are a good opportunity for the test of the method described in Section 4.1.

In the framework of the AURORA project (Cortesi et al., 2018) we simulated Sentinel 4 ozone vertical profile measurements

as they could be obtained by the Infrared Sounder operating in the thermal infrared on board the Meteosat Third Generation satellite (http://www.eumetsat.int/website/home/Satellites/FutureSatellites/MeteosatThirdGeneration/MTGDesign/). The Sentinel-4 and the Sentinel-5P observations will improve our ozone composition knowledge (Quesada-Ruiz et al., 2019) and the AURORA project is assessing the advantages offered by CDF in the exploitation of the data. The atmosphere used for the simulations is taken from the Modern Era-Retrospective analysis for Research and Applications version 2 (MERRA2)

reanalysis (Gelaro et al., 2017). The MERRA2 data are provided by the Global Modelling and Assimilation Office (GMAO) at NASA Goddard Space Flight Center. This reanalysis covers the modern era of remotely sensed data, from 1979 through the present. The data of a geostationary image, acquired on 1$^{\mathrm{st}}$ April 2012 in about one hour, were considered and of the available 423719 measurements only the 35594 measurements in clear sky have been simulated. A coincidence cell of 0,5° step of latitude and 0.625° step of longitude was chosen for the data fusion and a total of 1296 cells, where there are at least two

measurements that can be fused, are obtained. The time coincidence is in our case very short and is practically negligible.

The a priori profiles provided by the McPeters and Labow climatology (McPeters and Labow, 2012) are used for all fusing and fused profiles. The a priori CMs are obtained using the standard deviation of the McPeters and Labow climatology when its value is larger than 20% of the a priori profile and a value of 20% of the a priori profile in the other cases. The off diagonal elements are calculated considering a correlation length of 6 km. The correlation length provides an effective regularization

that reduces oscillations in the retrieved profiles and the value of 6 km is typically used for nadir ozone profile retrieval (Liu et al., 2010, Kroon et al., 2011, Miles et al., 2015).

The method described in Section 4.1 to determine the coincidence CMs for the fusion of these simulated data is used. We model $\mathbf{S}_{\mathrm{coin}}$ as $k\mathbf{S}_a$, that is we make the hypothesis that the variability of the true profiles is a fraction of that represented by the a priori CM, which describes the climatological variability of the true profiles on geographical regions that are larger than the

fusion cells.

In Fig. 1, we report the $k$ values given by Eq. (29) as a function of the number of fusing profiles, the $\Delta k$ errors are given by the colour scale. The second panel provides an enlargement of the first one for small values of $k$. From Fig. 1 we see that large values of $k$ are obtained when the number of fusing profiles is small and large errors are present. Since $k$ is a positive parameter,



the uncertainty in its determination manifests itself mainly with large positive values and a sufficient statistics is needed for a useful determination of $k$ and in our case the number of fusing profiles must be greater than 10.

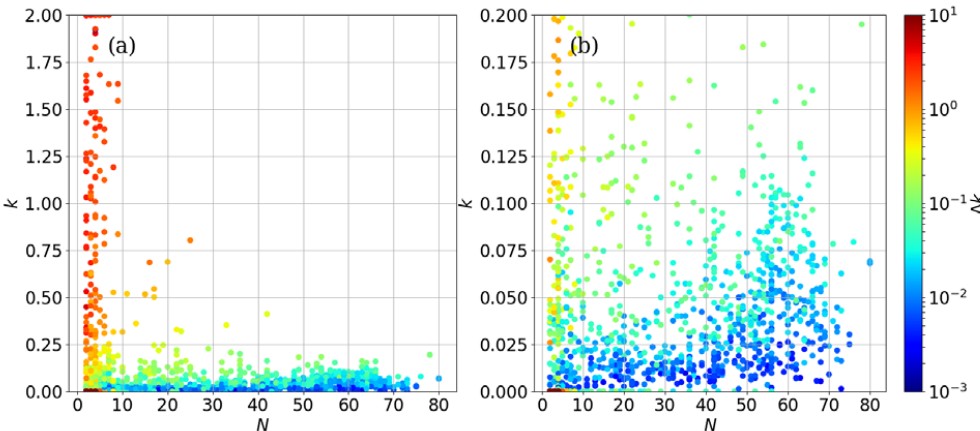

**Figure 1: (a) Values of the parameter $k$ as a function of the number of fusing profiles. (b) Enlargement of panel (a) for small values of $k$. The $\Delta k$ errors are reported in the colour scale.**

Increasing the number of fusing profiles the errors decrease and smaller values of $k$ are obtained, although the $\Delta k$ uncertainty, together with differences in the geographical variability, is still responsible for some dispersion of the $k$ values. When the number of fusing profiles is sufficient to produce reliable $k$ values we obtain $k$ values that are a fraction of the unity confirming that a small geographical variability, a fraction of the climatological variability, occurs within the cell chosen for the data fusion. In order to assess the entity of the obtained values it is important to notice that $k$ multiplies the CM and, accordingly, is proportional to the square of the geographical variability.

### 4.2.2 Results for a single cell with a large number of fusing profiles

As an example, we analyze the behavior of a cell with a large number of fusing profiles for which the $k$ value is well determined being significantly larger than the error $\Delta k$. We deal with a cell with 80 fusing profiles, for which, applying the method described in Section 4.1, we obtain $k = 0.068$ and $\Delta k = 0.014$. In this case, $\Delta k$ is about one fifth of the $k$ value.

The use of simulated data makes it possible to compare the results with the true quantities that we want to measure. In Fig. 2, we report the differences between three fused profiles, obtained with $k = 0.068$, with $k = 0$ and with the method used in the previous paper on the importance of coincidence errors (Ceccherini et al., 2018), and the true profile of the fusion, calculated as the mean of the true profiles corresponding to the fusing profiles. In the previous paper, an educated guess was made of the coincidence error and $S_{coin}$ equal to a matrix with the square of the 5% of the a priori profile on the diagonal elements and a correlation length of 6 km for the off diagonal elements was used. In the figure, also the errors and the numbers of DOFs of the three fused profiles are reported.

We see that the fused profile with $k = 0$ has large differences with respect to the true profile of the fusion, while the other two fused profiles have smaller and comparable differences. The errors are basically the same for all three fused profiles and the numbers of DOFs are about equal for $k = 0.068$ and for the method used in the previous paper, and slightly larger for $k = 0$. The importance of using a coincidence CM is confirmed because it provides a significant reduction of the differences with the true profile at the cost of a negligible reduction of the number of DOFs. The difference between the results obtained with the two coincidence CMs is small, but the method described in Section 4.1 provides a slightly better compromise between reproduction of the true profile and number of DOFs and, more important, is an objective determination based on a mathematical constraint.

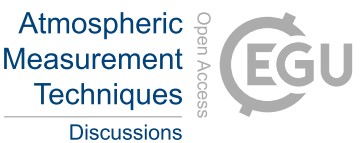



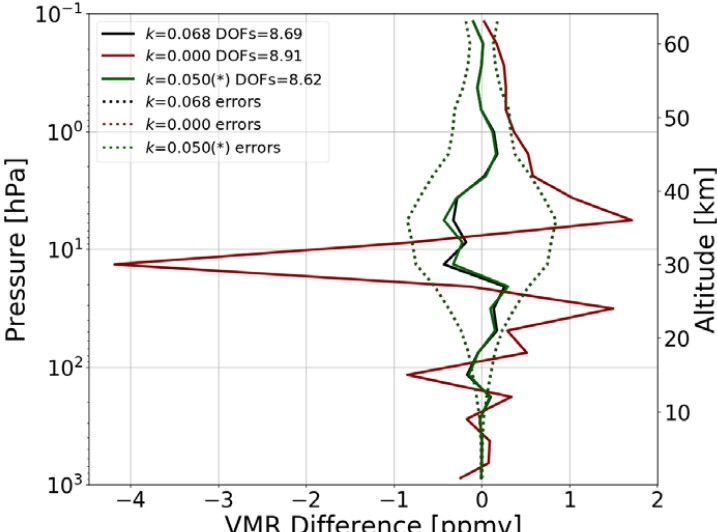

**Figure 2: Differences between the fused profiles obtained with *k* = 0.068 (black line), *k* = 0 (red line) and the method used in Ceccherini et al. (2018) (green line) (\* in this case *k* is applied to a CM built in a slightly different way, see text) and the mean of the true profiles related to the fusing profiles as a function of altitude (and pressure). The errors (dotted lines) and the numbers of DOFs of the three fused profiles are reported as well.**

In Fig. 3, we report the square root of the diagonal elements of $S_{coin}$ estimated by the method described in Section 4.1 (with its errors) and by the method used in the previous paper as a function of altitude (and pressure) and compare them with the standard deviation of the true profiles corresponding to the fusing profiles.

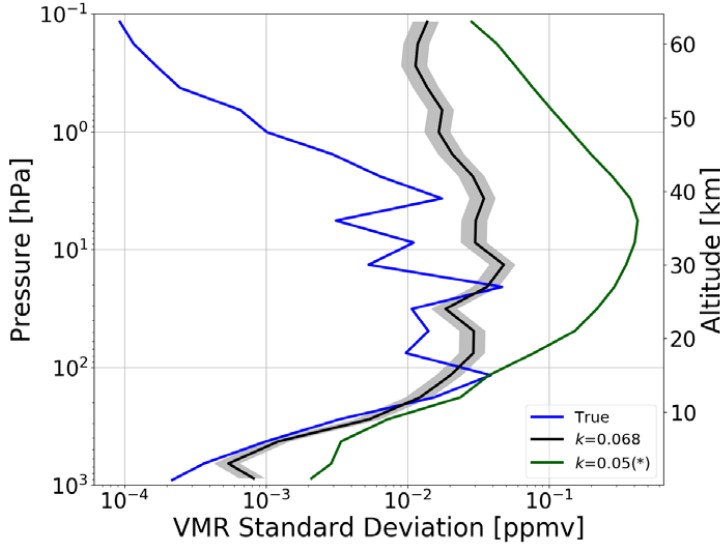

**Figure 3: Square root of the diagonal elements of $S_{coin}$ estimated by the method described in Section 4.1 (black line), with its errors (grey band around the black line), and by the educated guess used in Ceccherini et al. (2018) (green line) (\* in this case *k* is applied to a CM built in a slightly different way, see text) as a function of altitude (and pressure) compared with the standard deviation of the true profiles corresponding to the fusing profiles (blue line).**




We see that the method described in Section 4.1 is able to reproduce well the standard deviation of the true profiles up to about 30 km of altitude. Above 30 km this method overestimates the spread of the true profiles, probably because we assume $S_{coin}$ proportional to the a priori CM, which includes the day-night variability of ozone. This variability is instead absent in the fusing profiles because they belong to a single geostationary image that is acquired in one hour. The educated guess of $S_{coin}$

significantly overestimates the standard deviation of the true profiles below 8 km and above 15 km of altitude.

Within the limits posed by the fact that a single parameter is used for the estimate of a CM, the coincidence error determined with the constraint of the cost function is a very good representation of the real geographical variability, much better than that obtained with the educated guess (please note the logarithmic scale in Fig. 3), although the effect of this difference on the fusion process is very small, given the negligible consequences of overestimates of the coincidence error.

**4.2.3 Analysis of all fusion cells**

In order to evaluate the performances of the method described in Section 4.1 we introduce a quantifier $\beta$ equal to the root of the square sum of the relative differences between the fused profile and its true profile:

$$\beta = \sqrt{\sum_{i=1}^{n}\left(\frac{x_{fi} - x_{ti}}{x_{ti}}\right)^2} \, , \tag{31}$$

where $x_{fi}$ is the i-th component of the fused profile, $x_{ti}$ is the i-th component of the true profile of the fusion and $n$ is the number of levels of the fused profile. We calculated this quantifier for all the fusion cells.

In Fig. 4, we show the scatter plots of $\beta$ and of the number of DOFs (panel (a) and panel (b), respectively) of the fused profiles obtained with the $k$ values determined by the method described in Section 4.1 as a function of the same quantities obtained with $k$ equal to zero. The number of fusing profiles is reported in the colour scale. In the case of $\beta$, small values are preferred; in the case of number of DOFs, large values are preferred.

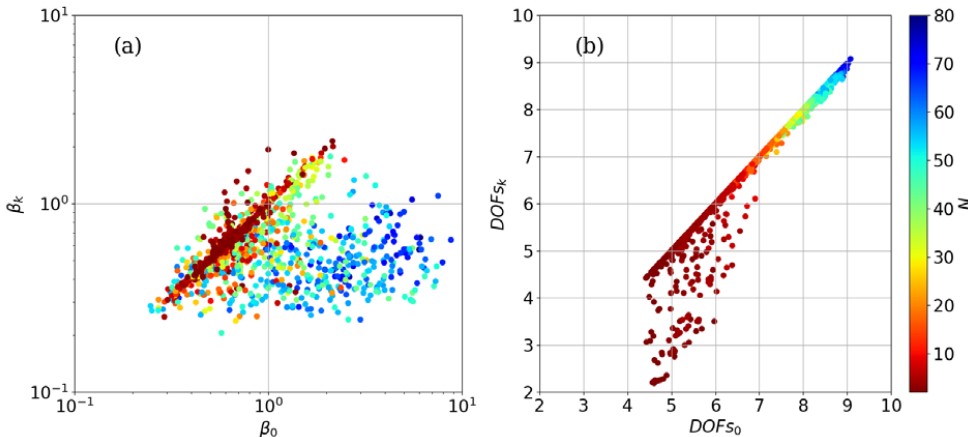

**Figure 4: (a) Scatter plot of $\beta_k$ as a function of $\beta_0$. The quantifier $\beta_k$ corresponds to the $k$ value determined by the method described in Section 4.1 and is reported in the y axis, while $\beta_0$ corresponds to $k$ equal to zero and is reported in the x axis. (b) Scatter plot of $DOFs_k$ as a function of $DOFs_0$. $DOFs_k$ is the number of DOFs of the fused profile obtained when $k$ is determined by the method described in Section 4.1 and is reported in the y axis, while $DOFs_0$ is the number of DOFs of the fused profile obtained when $k$ is equal to zero and is reported in the x axis. The number of fusing profiles is reported in the colour scale.**

From Fig. 4 we see that for large values of the number of fusing profiles in general the method described in Section 4.1 determines a significant reduction of $\beta$ with respect to the case of $k$ equal to zero, while the effect on the number of DOFs is negligible. In some cases for small values of the number of fusing profiles, we see that the use of the large value of $k$, erroneously determined by the method for the insufficient statistics, causes a significant reduction of the number of DOFs and




sometimes also an increase of $\beta$. A worse value of $\beta$ is obtained in a few cases also for cells that do not have a very small number of fusing profiles, however the loss observed in these cases is much smaller than the gain obtained in the much more numerous cells for which a reduction of $\beta$ is observed. The distribution of the colours in Fig. 4b clearly shows that the number of DOFs increases when the number of fusing profiles increases, confirming the improvement of information obtained with

the fusion of many profiles.

A complete evaluation of the performances of the method has to take into account both the ability to reproduce the true profile (represented by $\beta$) and the number of DOFs. For this reason, we define a new quantifier $\gamma$, equal to the ratio between $\beta$ and the number of DOFs, which takes into account both aspects:

$$\gamma = \frac{\beta}{DOFs} \ . \tag{32}$$

The quality of the fused profile improves when the value of $\gamma$ is reduced. In Fig. 5, we show the scatter plot of $\gamma$ of the fused

profiles obtained with the $k$ values determined by the method described in Section 4.1 as a function of $\gamma$ of the fused profiles obtained with $k$ equal to zero. If the number of fusing profiles is smaller than 10 the points are reported in red otherwise in blue.

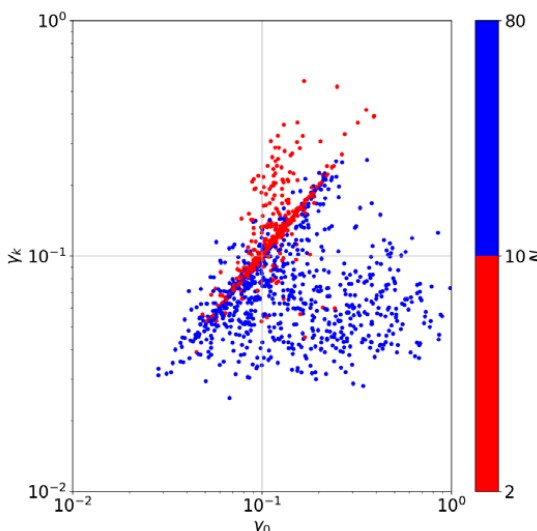

**Figure 5: Scatter plot of $\gamma_k$ as a function of $\gamma_0$. The quantifier $\gamma_k$ corresponds to the $k$ value determined by the method described in**
**Section 4.1 and is reported in the y axis, while $\gamma_0$ corresponds to $k$ equal to zero and is reported in the x axis. If the number of fusing profiles is smaller than 10 the points are reported in red otherwise in blue.**

From Fig. 5 we see that when the number of fusing profiles is larger than 10 in general the method described in Section 4.1 determines a reduction of $\gamma$, improving the quality of the fused profiles. Occasionally a worsening is observed, but improvements, in number and in entity, are overwhelmingly larger than worsenings. When the number of fusing profiles is

smaller than 10 the $k$ values determined by the method are affected by large errors (see Fig. 1) and values that are much larger than a reasonable expectation may be obtained. Therefore, it is not a surprise that in these cases the dominant effect is a degradation of the quality of the fused profiles with respect to the case of $k$ equal to zero. For this reason, the method described in Section 4.1, can only be used when either $k$ is determined with a small error or, similarly, the number of fusing profiles is sufficiently large. In the other cases, an educated guess should be used, possibly supported by the indications provided by the

results obtained in the cells with a large number of measurements.





**5 Conclusions**

The measurements that we wish to fuse often have some inconsistencies due to representations on different vertical grids, imperfect time and space coincidence and different forward model errors. In order to apply the CDF method to inconsistent measurements it is necessary to add to the measurement CM of each fusing profile a CM that qualifies these inconsistencies

as errors and prevents their use as erroneous features of the profile. Therefore, a realistic estimate of the inconsistency CM is required for effectual fused products. In this paper, we propose to use the statistical properties of the cost function distribution to improve the estimate of the inconsistency CM.

The expected value and the variance of the cost function distribution of the data fusion have been analytically determined for the first time. This allowed us to calculate the reduced cost function, which is bound to be equal to one within the variability

determined by its variance. Modelling the inconsistency CM with one parameter, we used the expected value of the reduced cost function as a constraint to tune the value of this parameter and the variance of the reduced cost function to assign an error to this value.

We applied this method to simulated measurements of ozone profiles obtained in the thermal infrared in the framework of the Sentinel 4 mission of the Copernicus programme. The results show that when the number of fusing profiles is small the values

of the parameter are affected by large errors, in particular they are almost completely undetermined if the number of fusing profiles is smaller than 10. For such small values of the number of fusing profiles, the method is not able to provide reliable values of the parameter and it is better to use an educated guess for the estimate of the inconsistency CM. On the other hand, when the number of fusing profiles is large enough the values of the parameter provided by the method are affected by small errors and the estimated coincidence CMs generally improve the performances of the CDF method, providing a significant

reduction of the differences between retrieved profile and true profile, with a negligible reduction of the number of DOFs.

*Data availability.* The data of the simulations presented in the paper are available upon request to the authors.

**Appendix A**

In this appendix we make the calculation of the variance of the cost function given in Eq. (24).

The variance is equal to:

$$\text{var}\left[ c^{\min}\left( \boldsymbol{\sigma}_i \right) \right] = \left\langle \left[ c^{\min}\left( \boldsymbol{\sigma}_i \right) - \left\langle c^{\min}\left( \boldsymbol{\sigma}_i \right) \right\rangle \right]^2 \right\rangle . \tag{A1}$$

Substituting in Eq. (A1) the expression of $c^{\min}\left( \boldsymbol{\sigma}_i \right)$ given by Eq. (17), we obtain:

$$\text{var}\left[ c^{\min}\left( \boldsymbol{\sigma}_i \right) \right] = \left\langle \left[ c^{\min}_0 + c^{\min}_1\left( \boldsymbol{\sigma}_i \right) + c^{\min}_2\left( \boldsymbol{\sigma}_i \right) - \left\langle c^{\min}_0 \right\rangle - \left\langle c^{\min}_1\left( \boldsymbol{\sigma}_i \right) \right\rangle - \left\langle c^{\min}_2\left( \boldsymbol{\sigma}_i \right) \right\rangle \right]^2 \right\rangle =$$
$$= \left\langle \left[ c^{\min}_1\left( \boldsymbol{\sigma}_i \right) + c^{\min}_2\left( \boldsymbol{\sigma}_i \right) - \left\langle c^{\min}_2\left( \boldsymbol{\sigma}_i \right) \right\rangle \right]^2 \right\rangle = \left\langle \left[ c^{\min}_1\left( \boldsymbol{\sigma}_i \right) \right]^2 \right\rangle + \left\langle \left[ c^{\min}_2\left( \boldsymbol{\sigma}_i \right) \right]^2 \right\rangle - \left\langle c^{\min}_2\left( \boldsymbol{\sigma}_i \right) \right\rangle^2 \tag{A2}$$

where we used $<c^{\min}_0>=c^{\min}_0$, $<c^{\min}_1(\boldsymbol{\sigma}_i)>=0$ and $<c^{\min}_1(\boldsymbol{\sigma}_i)c^{\min}_2(\boldsymbol{\sigma}_i)>=0$ because the product $c^{\min}_1(\boldsymbol{\sigma}_i)c^{\min}_2(\boldsymbol{\sigma}_i)$ is cubic in the errors and, therefore, its expected value is zero as a consequence of the symmetry of the normal distribution.

Using Eq. (19), for the first term of Eq. (A2) we obtain:

$$\left\langle \left[ c^{\min}_1\left( \boldsymbol{\sigma}_i \right) \right]^2 \right\rangle = 4 \left\langle \boldsymbol{\sigma}_f^T \mathbf{S}_a^{-1}\left( \mathbf{x}_t - \mathbf{x}_a \right)\left( \mathbf{x}_t - \mathbf{x}_a \right)^T \mathbf{S}_a^{-1} \boldsymbol{\sigma}_f \right\rangle =$$
$$= 4tr\left[ \mathbf{S}_f \mathbf{S}_a^{-1}\left( \mathbf{x}_t - \mathbf{x}_a \right)\left( \mathbf{x}_t - \mathbf{x}_a \right)^T \mathbf{S}_a^{-1} \right] = 4tr\left[ \left( \mathbf{x}_t - \mathbf{x}_a \right)\left( \mathbf{x}_t - \mathbf{x}_a \right)^T \mathbf{S}_a^{-1} \mathbf{A}_f \left( \mathbf{I} - \mathbf{A}_f \right) \right] \tag{A3}$$

where we have used the relation $\mathbf{S}_f \mathbf{S}_a^{-1} = \mathbf{A}_f \left( \mathbf{I} - \mathbf{A}_f \right)$ that comes from Eqs. (6) and (7).



Using Eq. (20), for the second term of Eq. (A2) we obtain:

$$\left\langle\left[c^{\min}{}_{2}\left(\boldsymbol{\sigma}_{i}\right)\right]^{2}\right\rangle=\left\langle\left[\sum_{i=1}^{N}\boldsymbol{\sigma}_{i}^{T}\mathbf{S}_{i}^{-1}\boldsymbol{\sigma}_{i}\right]^{2}\right\rangle+\left\langle\left[\boldsymbol{\sigma}_{f}^{T}\left(\mathbf{F}+\mathbf{S}_{a}^{-1}\right)\boldsymbol{\sigma}_{f}\right]^{2}\right\rangle-2\left\langle\left[\sum_{i=1}^{N}\boldsymbol{\sigma}_{i}^{T}\mathbf{S}_{i}^{-1}\boldsymbol{\sigma}_{i}\boldsymbol{\sigma}_{f}^{T}\left(\mathbf{F}+\mathbf{S}_{a}^{-1}\right)\boldsymbol{\sigma}_{f}\right]^{2}\right\rangle. \tag{A4}$$

Some further elaboration is needed to evaluate these three terms:

$$\left\langle\left[\sum_{i=1}^{N}\boldsymbol{\sigma}_{i}^{T}\mathbf{S}_{i}^{-1}\boldsymbol{\sigma}_{i}\right]^{2}\right\rangle=\sum_{i=1}^{N}\left\langle\boldsymbol{\sigma}_{i}^{T}\mathbf{S}_{i}^{-1}\boldsymbol{\sigma}_{i}\boldsymbol{\sigma}_{i}^{T}\mathbf{S}_{i}^{-1}\boldsymbol{\sigma}_{i}\right\rangle+\sum_{\substack{i,k=1\\i\neq k}}^{N}\left\langle\boldsymbol{\sigma}_{i}^{T}\mathbf{S}_{i}^{-1}\boldsymbol{\sigma}_{i}\right\rangle\left\langle\boldsymbol{\sigma}_{k}^{T}\mathbf{S}_{k}^{-1}\boldsymbol{\sigma}_{k}\right\rangle=$$

$$=2\sum_{i=1}^{N}tr\left(\mathbf{S}_{i}^{-1}\mathbf{S}_{i}\mathbf{S}_{i}^{-1}\mathbf{S}_{i}\right)+\sum_{i=1}^{N}\left[tr\left(\mathbf{S}_{i}^{-1}\mathbf{S}_{i}\right)\right]^{2}+\sum_{\substack{i,k=1\\i\neq k}}^{N}tr\left(\mathbf{S}_{i}\mathbf{S}_{i}^{-1}\right)tr\left(\mathbf{S}_{k}\mathbf{S}_{k}^{-1}\right)= \tag{A5}$$

$$=2\sum_{i=1}^{N}n_{i}+\sum_{i,k=1}^{N}n_{i}n_{k}=\sum_{i=1}^{N}n_{i}\left(2+\sum_{k=1}^{N}n_{k}\right)$$

$$\left\langle\left[\boldsymbol{\sigma}_{f}^{T}\left(\mathbf{F}+\mathbf{S}_{a}^{-1}\right)\boldsymbol{\sigma}_{f}\right]^{2}\right\rangle=2tr\left[\left(\mathbf{F}+\mathbf{S}_{a}^{-1}\right)\mathbf{S}_{f}\left(\mathbf{F}+\mathbf{S}_{a}^{-1}\right)\mathbf{S}_{f}\right]+\left\{tr\left[\left(\mathbf{F}+\mathbf{S}_{a}^{-1}\right)\mathbf{S}_{f}\right]\right\}^{2}=$$

$$=2tr\left(\mathbf{A}_{f}^{2}\right)+\left[tr\left(\mathbf{A}_{f}\right)\right]^{2} \tag{A6}$$

and

$$-2\left\langle\left[\sum_{i=1}^{N}\boldsymbol{\sigma}_{i}^{T}\mathbf{S}_{i}^{-1}\boldsymbol{\sigma}_{i}\boldsymbol{\sigma}_{f}^{T}\left(\mathbf{F}+\mathbf{S}_{a}^{-1}\right)\boldsymbol{\sigma}_{f}\right]^{2}\right\rangle=-2\sum_{i=1}^{N}\left\langle\boldsymbol{\sigma}_{i}^{T}\mathbf{S}_{i}^{-1}\boldsymbol{\sigma}_{i}\boldsymbol{\sigma}_{i}^{T}\mathbf{S}_{i}^{-1}\mathbf{A}_{i}\left(\mathbf{F}+\mathbf{S}_{a}^{-1}\right)^{-1}\mathbf{A}_{i}^{T}\mathbf{S}_{i}^{-1}\boldsymbol{\sigma}_{i}\right\rangle-$$

$$-2\sum_{\substack{i,k=1\\i\neq k}}^{N}\left\langle\boldsymbol{\sigma}_{i}^{T}\mathbf{S}_{i}^{-1}\boldsymbol{\sigma}_{i}\right\rangle\left\langle\boldsymbol{\sigma}_{k}^{T}\mathbf{S}_{k}^{-1}\mathbf{A}_{k}\left(\mathbf{F}+\mathbf{S}_{a}^{-1}\right)^{-1}\mathbf{A}_{k}^{T}\mathbf{S}_{k}^{-1}\boldsymbol{\sigma}_{k}\right\rangle= \tag{A7}$$

$$=-2\sum_{i=1}^{N}2tr\left[\mathbf{S}_{i}^{-1}\mathbf{A}_{i}\left(\mathbf{F}+\mathbf{S}_{a}^{-1}\right)^{-1}\mathbf{A}_{i}^{T}\right]-2\sum_{i=1}^{N}tr\left(\mathbf{S}_{i}^{-1}\mathbf{S}_{i}\right)tr\left[\mathbf{S}_{i}^{-1}\mathbf{A}_{i}\left(\mathbf{F}+\mathbf{S}_{a}^{-1}\right)^{-1}\mathbf{A}_{i}^{T}\right]-$$

$$-2\sum_{\substack{i,k=1\\i\neq k}}^{N}tr\left(\mathbf{S}_{i}^{-1}\mathbf{S}_{i}\right)tr\left[\mathbf{S}_{k}^{-1}\mathbf{A}_{k}\left(\mathbf{F}+\mathbf{S}_{a}^{-1}\right)^{-1}\mathbf{A}_{k}^{T}\right]=-2tr\left(\mathbf{A}_{f}\right)\left(2+\sum_{i=1}^{N}n_{i}\right)$$

where we have used the formula for the expected value of the quartic form given in Petersen and Pedersen (2012), Eqs. (5-7) and Eq. (15).

5   The third term of Eq. (A2) is given by Eq. (22).

From Eq. (A2), using Eq. (22) and Eqs. (A3-A7), we obtain the expression of the variance of the cost function:

$$\mathrm{var}\left[c^{\min}\left(\boldsymbol{\sigma}_{i}\right)\right]=2\sum_{i=1}^{N}n_{i}-4tr\left(\mathbf{A}_{f}\right)+2tr\left(\mathbf{A}_{f}^{2}\right)+4tr\left[\left(\mathbf{x}_{t}-\mathbf{x}_{a}\right)\left(\mathbf{x}_{t}-\mathbf{x}_{a}\right)^{T}\mathbf{S}_{a}^{-1}\mathbf{A}_{f}\left(\mathbf{I}-\mathbf{A}_{f}\right)\right]. \tag{A8}$$

*Author contributions.* SC calculated the expected value and the variance of the cost function and wrote the draft version of the paper. NZ wrote the Python code of the complete data fusion and applied the procedure to determine the coincidence

10   covariance matrix to the ozone simulated measurements. BC suggested the idea to use the cost function to determine the inconsistency covariance matrices and contributed to the interpretation of the results. SDB performed the simulation of the ozone measurements. UC and CT are, respectively, the principal investigator and the project manager of the AURORA project and coordinated the activity of the project. All the authors revised the manuscript.

15   *Competing interests.* The authors declare that they have no conflict of interest.





*Acknowledgments.* The results presented in this paper arise from research activities conducted in the framework of the AURORA project (http://www.aurora-copernicus.eu/) supported by the Horizon 2020 research and innovation programme of the European Union (Call: H2020-EO-2015; Topic: EO-2-2015) under Grant Agreement N. 687428.

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
