# Peer review of "The cost function of the data fusion process and its application"

_Atmospheric Measurement Techniques, 2019_

## Referee Comment (RC1) · Anonymous Referee #2 · 8 Mar 2019

The paper by Checcherini et al. covers a topic that fits well within the scope of AMT. It seems scientifically sound to me in most of its parts. The paper is well written, and the text between the equations guides the reader safely through the maths. The notation chosen in some places seems a bit odd to me. I suggest publication after clarification of the following issues.

p1 l18 ff: The first paragraph of the intro is very informative and clear. If helps to pick up the thread. Congratulations!

p1 l26: To me it is counter-intuitive that the equivalence holds also in nonlinear cases. Does this statement need a qualification "equivalent ... in the linear case, and

approximately equivalent in the moderately non-linear case"? I have mentioned this already in the access review and since the authors did not change anything I suspect that this has been done by intention and I may have missed the main argument. Just to make sure that things are correct, I suggest to either add "within linear theory" to the text, or to provide (in the rebuttal, not in the paper) a short justification why this statement holds also in the non-linear case.

p1 l35: It seems to me to be ambiguous what the interpolation error shall represent. Is it (a) that fine structures of the true profile are not represented by the profiles sampled on a coarse grid, and that interpolation to a finer grid will not recover these fine structures? If so, I would object that these issues are already covered by the averaging kernel of the coarse grid representation, and it is not clear to me why an extra treatment is needed. Or (b) is the source of this inconsistency an inconsistency of different interpolation methods. Here it is important that data points of a profile retrieval do not represent a single point in the atmosphere but that the value retrieved at this point depends on the profile assumed for the entire layer between two levels. This is because the forward solution of the radiative transfer equation is basically an integration along the line of sight, and thus, assumptions of the atmospheric state between the levels are used (regardless if an Curtis-Godson approach or a direct integration method is chosen). For this purpose, the forward model typically uses an internal interpolation in the retrieval. The solution of the retrieval based on this forward model is conditional to this internal interpolation or interpolation scheme or assumption. Assuming a different behavior between the layers would result in different retrieved values at the model levels. I think it needs to be elaborated why an interpolation uncertainty is introduced if an interpolation scheme is used which is consistent with that in the forward model. Or do the authors assume that the interpolation of retrieved profiles is made with another interpolation scheme than that used in the forward model?

[Figure]

p2 l20: Here it is not clear what the term 'error' means. Is it a statistical estimate or is it the actual difference between the actual retrieved value and the true (or expectation) value? In other words, do the sigmas represent a statistic or an instantiation (or realization) of it? Normally, the symbol $\sigma$ is used for a statistic, namely the standard deviation. But this seems to make no sense in this context because the sigmas themselves are arguments of the expectation value function. If the errors (sigmas) are understood to be actual differences, then the paragraph makes sense to me but I challenge the adequacy of the notation. The symbol $\sigma$ is, in the context of error analysis, always used for a statistic, namely, a standard deviation, and it took me multiple readings of this paragraph to find out that the authors seem to mean something else. Our alphabet has so many letters available, each of them as capital and small character, and we have multiple alphabets available (Latin, Greece, ...). For the actual differences, d or delta would be an option but there are many others. Why choosing without need a notation which will almost certainly lead the reader astray?

p2 Eq 3: Here it becomes clear that sigma is not a standard deviation but a single instantiation of noise mapped from the signal space into the atmospheric state space. I consider this very unfortunate and misleading because sigma is usually reserved for the standard deviation; see my comment above.

p2 Eq 4: I consider $\mathbf{F}$ as an unfortunate choice of symbol for the Fisher information matrix, because in Rodgers' notation this symbol stands for the (vectorial) forward model. Since such a formal paper, even without notation-related complication, is hard enough to digest, I think, one could make the reader's life easier by choosing another letter in order to avoid confusion.

p3 Eq 11: According to p 2 line 25, $\mathbf{S}_a$ seems the a priori covariance represented on the grid on which the retrieval is performed. It does not contain any variability on any finer scale (see von Clarmann, AMT 7, 3023-3034, 2014 for similar considerations regarding the role of $\mathbf{S}_a$ in the calculation of the smoothing error.) The $\mathbf{C}^{(f)}$ matrix involves a finer grid. Is thus the $\mathbf{S}_a$ matrix, which is evaluated on the retrieval grid, sufficient for the

purpose of Eq. 11? This is not immediately intuitive and needs some discussion. Or have you tacitly redefined $S_a$? If so, this should be stated explicitly. This equation seems to give the answer to my question about the meaning of the interpolation error (p1 l35), and the interpretation of the interpolation error seems to be option (a). Given the problems caused by the grid-dependence of $S_a$, wouldn't it be safer to dismiss the idea to characterize the interplation effect as a covariance matrix and to provide an averaging kernel of the resulting fused profile which includes also the interpolation effects instead?

Section 3 ff: I cannot guarantee that all the manipulations of equations are flawless but the general rationale behind seems compelling to me.

Technical issues: Subscripts of matrices which are neither variables nor indices but denote simply the name of the matrix should not be printed italic but roman (e.g. the a in $S_a$, the 'int, coin, other' in the $S_{i,\text{int}}$, $S_{i,\text{coin}}$, $S_{i,\text{other}}$, respectively, etc).

The similarity index appears to be quite high but I consider the detected similarities as insignificant. The index seems to be very sensitive to standard formulations that everybody uses.

---

## Referee Comment (RC2) · Anonymous Referee #1 · 9 Mar 2019

This paper is a continuation of the series of papers authored by Ceccherini et al. over the last few years. It continues the good work describing the features of the complete data fusion (CDF) method. As such, it is worthy of publication, provided the authors address the following general and specific comments.

The main general comment is the suggestion that the authors transfer even more of the technical discussions (e.g., in Sect. 3), including representation of equations, to an appendix (I have not checked all the mathematical workings, but they seem correct to me). Whether this should take place is a decision between the authors and the editor.

Generally, the authors have done a good job of writing the paper. I have a minor correction:

P. 10

L. 2: probably -> likely.

---

## Author Comment (AC1) · 30 Apr 2019

We thank the reviewer for the useful comments. In the following, we answer the specific comments (included in "**boldface**" for clarity) and, whenever required, we describe the related changes implemented in the revised manuscript. Page and line numbers indicated refer to the original version of the paper published on AMTD.

**Anonymous Referee #1**

**Review**

**This paper is a continuation of the series of papers authored by Ceccherini et al. over the last few years. It continues the good work describing the features of the complete data fusion (CDF) method. As such, it is worthy of publication, provided the authors address the following general and specific comments.**

**The main general comment is the suggestion that the authors transfer even more of the technical discussions (e.g., in Sect. 3), including representation of equations, to an appendix (I have not checked all the mathematical workings, but they seem correct to me). Whether this should take place is a decision between the authors and the editor.**

We have already moved the lengthy calculation of the cost function variance in an Appendix, now we have difficulty in moving the remaining mathematics because it is an important part of the paper and involves definitions that we believe the reader prefers to find in the main text. Nevertheless, we are open to suggestions by the editor.

**Generally, the authors have done a good job of writing the paper. I have a minor correction:**
**P. 10**
**L. 2: probably -> likely.**
In the revised version of the paper we made the correction suggested by the reviewer.

---

## Author Comment (AC2)

We thank the reviewer for the useful comments. In the following, we answer the specific comments (included in "**boldface**" for clarity) and, whenever required, we describe the related changes implemented in the revised manuscript. Page and line numbers indicated refer to the original version of the paper published on AMTD.

**Anonymous Referee #2**

**Review**

**The paper by Ceccherini et al. covers a topic that fits well within the scope of AMT. It seems scientifically sound to me in most of its parts. The paper is well written, and the text between the equations guides the reader safely through the maths. The notation chosen in some places seems a bit odd to me. I suggest publication after clarification of the following issues.**

**p1 l18 ff: The first paragraph of the intro is very informative and clear. It helps to pick up the thread. Congratulations!**
We thank the reviewer.

**p1 l26: To me it is counter-intuitive that the equivalence holds also in nonlinear cases. Does this statement need a qualification "equivalent ... in the linear case, and approximately equivalent in the moderately non-linear case"? I have mentioned this already in the access review and since the authors did not change anything I suspect that this has been done by intention and I may have missed the main argument. Just to make sure that things are correct, I suggest to either add "within linear theory" to the text, or to provide (in the rebuttal, not in the paper) a short justification why this statement holds also in the non-linear case.**
We could not find any reference to this point in the access review, otherwise we would have addressed this point before. In the point indicated by the reviewer (p1 l26) we are speaking about the equivalence of the CDF method with the measurement space solution data fusion method. The rigorous proof of this equivalence is given in Ceccherini (2016). Because both methods use a linearization in a range close to the retrieval results the two methods are equivalent.
Maybe the reviewer is referring to one line above (p1 l25), where we are talking about the equivalence of quality of the products of the CDF and of the simultaneous retrieval (which in general is a non-linear process). This equivalence is addressed in Ceccherini et al. (2015) and the rigorous proof is given under the assumption that a linear approximation can be applied to the forward model of each measurement in the range of variability between the solution of the single retrieval and that of the simultaneous retrieval. Therefore, we agree with the reviewer that a clarification is useful and in the revised version of the paper we have added a sentence accordingly.

**p1 l35: It seems to me to be ambiguous what the interpolation error shall represent. Is it (a) that fine structures of the true profile are not represented by the profiles sampled on a coarse grid, and that interpolation to a finer grid will not recover these fine structures? If so, I would object that these issues are already covered by the averaging kernel of the coarse grid representation, and it is not clear to me why an extra treatment is needed. Or (b) is the source of this inconsistency an inconsistency of different interpolation methods. Here it is important that data points of a profile retrieval do not represent a single point in the atmosphere but that the value retrieved at this point depends on the profile assumed for the entire layer between two levels. This is because the forward solution of the radiative transfer equation is basically an integration along the line of sight, and thus, assumptions of the atmospheric state between the**

**levels are used (regardless if an Curtis-Godson approach or a direct integration method is chosen). For this purpose, the forward model typically uses an internal interpolation in the retrieval. The solution of the retrieval based on this forward model is conditional to this internal interpolation or interpolation scheme or assumption. Assuming a different behavior between the layers would result in different retrieved values at the model levels. I think it needs to be elaborated why an interpolation uncertainty is introduced if an interpolation scheme is used which is consistent with that in the forward model. Or do the authors assume that the interpolation of retrieved profiles is made with another interpolation scheme than that used in the forward model?**

The interpolation error and its importance in data fusion is deeply analyzed in Ceccherini et al. (2018). The interpolation error here discussed is neither case (a), a resolution limit of the retrieved product, nor case (b), inconsistency of interpolation laws between forward model and subsequent operations. It is the effect of the resampling of the retrieved product that is made for the fusion of profiles with different retrieval grids. When the vertical grids of the fusing profiles differ from the fusion grid it is necessary to perform an interpolation of the rows of the AKMs on the fusion grid. Since the interpolated AKMs are only an approximation of the real AKMs on the fusion grid, we have to take into account this approximation including a new error component that we call interpolation error.

**p2 l20: Here it is not clear what the term 'error' means. Is it a statistical estimate or is it the actual difference between the actual retrieved value and the true (or expectation) value? In other words, do the sigmas represent a statistic or an instantiation (or realization) of it? Normally, the symbol σ is used for a statistic, namely the standard deviation. But this seems to make no sense in this context because the sigmas themselves are arguments of the expectation value function. If the errors (sigmas) are understood to be actual differences, then the paragraph makes sense to me but I challenge the adequacy of the notation. The symbol σ is, in the context of error analysis, always used for a statistic, namely, a standard deviation, and it took me multiple readings of this paragraph to find out that the authors seem to mean something else. Our alphabet has so many letters available, each of them as capital and small character, and we have multiple alphabets available (Latin, Greece, ...). For the actual differences, d or delta would be an option but there are many others. Why choosing without need a notation which will almost certainly lead the reader astray?**

The term 'error' is meant as a realization of it, not with a statistical meaning. The reviewer is right stating that the choice of the symbol sigma is not the most appropriate, because very often it is used to indicate the standard deviation of a statistical variable. However, since in the previous papers describing the CDF we have always used the symbol sigma as in this paper (see Ceccherini et al. 2015, Ceccherini 2016 and Ceccherini et al 2018), for comfort of the reader interested in CDF we think that it is better to maintain the notation used in the previous papers. Furthermore, as the reviewer states the misunderstanding of sigma as a standard deviation is not possible, because when it is introduced it appears as argument of an expectation value.

In the revised version of the paper, in order to avoid the reader astray, we have added the adjective "actual" to the word "errors".

**p2 Eq 3: Here it becomes clear that sigma is not a standard deviation but a single instantiation of noise mapped from the signal space into the atmospheric state space. I consider this very unfortunate and misleading because sigma is usually reserved for the standard deviation; see my comment above.**

See the answer to the previous comment.

**p2 Eq 4: I consider F as an unfortunate choice of symbol for the Fisher information matrix, because in Rodgers' notation this symbol stands for the (vectorial) forward model. Since such**

**a formal paper, even without notation-related complication, is hard enough to digest, I think, one could make the reader's life easier by choosing another letter in order to avoid confusion.**
The forward model is a vector valued function of the state and in the Rodgers' book it is indicated as $\mathbf{F}(\mathbf{x})$, while $\mathbf{F}$ alone is never used. On the other hand, in our paper, where the forward model is not used it is difficult to confuse it with the Fisher matrix. Furthermore, as specified above we prefer to maintain coherence with our previous papers such as Ceccherini et al. 2012 and Ceccherini et al. 2016, where the Fisher matrix is indicated as $\mathbf{F}$.

**p3 Eq 11: According to p 2 line 25, Sa seems the a priori covariance represented on the grid on which the retrieval is performed. It does not contain any variability on any finer scale (see von Clarmann, AMT 7, 3023-3034, 2014 for similar considerations regarding the role of $S_a$ in the calculation of the smoothing error.) The $C^{(f)}$ matrix involves a finer grid. Is thus the $S_a$ matrix, which is evaluated on the retrieval grid, sufficient for the purpose of Eq. 11? This is not immediately intuitive and needs some discussion. Or have you tacitly redefined Sa? If so, this should be stated explicitly. This equation seems to give the answer to my question about the meaning of the interpolation error (p1 l35), and the interpretation of the interpolation error seems to be option (a). Given the problems caused by the grid-dependence of $S_a$, wouldn't it be safer to dismiss the idea to characterize the interpolation effect as a covariance matrix and to provide an averaging kernel of the resulting fused profile which includes also the interpolation effects instead?**
The reviewer is right that a clarification on the grid on which $\mathbf{S}_a$ is represented is necessary. Eq. (1) is related to the case that all the fusing profiles are on a same vertical grid and, therefore, $\mathbf{S}_a$ is represented on that single grid. In Eq. (11), instead, we make the hypothesis that the fusing profiles and the fused profile are represented on different vertical grids and $\mathbf{S}_a$ is defined on the grid that includes all the levels of the fusion grid (*f*) and of the *N* grids (*i*).
In the revised version of the paper, we have added a sentence after Eq. (11) that clarifies this point.
Regarding the idea of the reviewer to include the interpolation effect in the averaging kernel of the resulting fused profile rather than in the covariance matrix, we think that it could be an interesting alternative, but we are not aware of any treatment of this kind in the literature.

**Section 3 ff: I cannot guarantee that all the manipulations of equations are flawless but the general rationale behind seems compelling to me.**

**Technical issues: Subscripts of matrices which are neither variables nor indices but denote simply the name of the matrix should not be printed italic but roman (e.g. the a in Sa, the 'int, coin, other' in the $S_{i,int}$, $S_{i,coin}$, $S_{i,other}$, respectively, etc).**
According to a similar recommendation made in the access review, we have already modified the style of $\mathbf{S}_{i,\text{other}}$, $\mathbf{S}_{i,\text{coin}}$, $\mathbf{S}_{\text{coin}}$ and $\mathbf{S}_{i,\text{FM}}$. In $\mathbf{S}_{i,\text{int}}$ "int" was already not italic in the original version of the paper. Regarding $\mathbf{S}_a$ we have left "a" in italic because this is done in the Rodgers' book and in all our previous papers.

**The similarity index appears to be quite high but I consider the detected similarities as insignificant. The index seems to be very sensitive to standard formulations that everybody uses.**

---

## Author Response (AR2)

Dear Dr. Butz,

actually the sigma_i is not equal to the departures between retrieved and true vertical profiles but it contains only the component due to the errors in the observations (the radiances). It does not contain the smoothing error component that is the component due to the use of a constraint in the retrieval.

Therefore, we have reworded the sentence in the following way:

"where the vector sigma_i contains the error due to the propagation of the observation noise through the retrieval process (and differs from the total error, equal to the difference between true and retrieved vertical profiles, by the smoothing error due to the use of a constraint in the retrieval)."

Best regards

Simone Ceccherini